# Ag/Au Bimetallic Nanoparticles Trigger Different Cell Death Pathways and Affect Damage Associated Molecular Pattern Release in Human Cell Lines

**DOI:** 10.3390/cancers14061546

**Published:** 2022-03-17

**Authors:** Hector Katifelis, Maria-Paraskevi Nikou, Iuliia Mukha, Nadiia Vityuk, Nefeli Lagopati, Christina Piperi, Ammad Ahmad Farooqi, Natassa Pippa, Efstathios P. Efstathopoulos, Maria Gazouli

**Affiliations:** 1Laboratory of Biology, Department of Basic Medical Sciences, Medical School, National and Kapodistrian University of Athens, 11527 Athens, Greece; katifel@med.uoa.gr (H.K.); mariaparaskevinikou@gmail.com (M.-P.N.); 2Chuiko Institute of Surface Chemistry, National Academy of Sciences of Ukraine, 01030 Kyiv, Ukraine; mukha@isc.gov.ua (I.M.); nvityuk@gmail.com (N.V.); 3Molecular Carcinogenesis Group, Department of Histology and Embryology, School of Medicine, National and Kapodistrian University of Athens, 11527 Athens, Greece; nlagopati@med.uoa.gr; 4Biomedical Research Foundation, Academy of Athens, 11527 Athens, Greece; 5Laboratory of Biological Chemistry, Department of Basic Medical Sciences, Medical School, National and Kapodistrian University of Athens, 11527 Athens, Greece; cpiperi@med.uoa.gr; 6Institute of Biomedical and Genetic Engineering (IBGE), Islamabad 44000, Pakistan; farooqiammadahmad@gmail.com; 7Section of Pharmaceutical Technology, Department of Pharmacy, School of Health Sciences, National and Kapodistrian University of Athens, 15771 Athens, Greece; natpippa@pharm.uoa.gr; 82nd Department of Radiology, Medical Physics Unit, Medical School, National and Kapodistrian University of Athens, Attikon University Hospital, 12462 Athens, Greece; stathise@med.uoa.gr; 9School of Science and Technology, Hellenic Open University, 26335 Patra, Greece

**Keywords:** bimetallic nanoparticles, apoptosis, pyroptosis, necroptosis, gold, silver

## Abstract

**Simple Summary:**

Apoptosis is the goal of several therapeutic strategies for cancer. However, the apoptotic pathway is not always functional in many cancers and thus, alternative ways to destroy cancer cells are required. In this context, we investigated whether nanoparticles composed of a gold and silver alloy (AgAu NPs) can induce other programmed cell death pathways. These include necroptosis and pyroptosis, while their effects on the release of molecules that serve as danger signals, the damage associated molecular patterns (DAMPs) were also investigated. Our findings suggest that MDA-MB-231 cells, one of the cancer cell lines tested, experience mixed cell death (several cell death pathways are activated), while a second cell line, HCT116 cells, releases DAMPS. This is important, since necroptosis and pyroptosis have promising anticancer effects, while DAMPs trigger inflammation and current knowledge suggests a rather beneficial role in cancer.

**Abstract:**

Apoptosis induction is a common therapeutic approach. However, many cancer cells are resistant to apoptotic death and alternative cell death pathways including pyroptosis and necroptosis need to be triggered. At the same time, danger signals that include HMGB1 and HSP70 can be secreted/released by damaged cancer cells that boost antitumor immunity. We studied the cytotoxic effects of AgAu NPs, Ag NPs and Au NPs with regard to the programmed cell death (apoptosis, necroptosis, pyroptosis) and the secretion/release of HSP70 and HMGB1. Cancer cell lines were incubated with 30, 40 and 50 μg/mL of AgAu NPs, Ag NPs and Au NPs. Cytotoxicity was estimated using the MTS assay, and mRNA fold change of *CASP1*, *CASP3*, *BCL-2*, *ZPB1*, *HMGB1*, *HSP70*, *CXCL8*, *CSF1*, *CCL20*, *NLRP3*, *IL-1β* and *IL-18* was used to investigate the associated programmed cell death. Extracellular levels of HMGB1 and IL-1β were investigated using the ELISA technique. The nanoparticles showed a dose dependent toxicity. Pyroptosis was triggered for LNCaP and MDA-MB-231 cells, and necroptosis for MDA-MB-231 cells. HCT116 cells experience apoptotic death and show increased levels of extracellular HMGB1. Our results suggest that in a manner dependent of the cellular microenvironment, AgAu NPs trigger mixed programmed cell death in P53 deficient MDA-MB-231 cells, while they also trigger IL-1β release in MDA-MB-231 and LNCaP cells and release of HMGB1 in HCT116 cells.

## 1. Introduction

Inducing apoptosis is a therapeutic strategy for malignancies that has been tested and studied thoroughly with satisfactory results [1]. However, several cancer types are apoptosis-resistant and thus, alternative programmed cell death pathways (PCD pathways) need to be sought for the therapeutic options to have the required efficiency [2].

These pathways are promising for anti-cancer approaches due to their unique characteristics. Pyroptosis, a lytic type of cell death has been shown to inhibit tumor growth [3] and was initially linked to infection. Its pathway consists of the activation of caspase 1 (CASP1), which cleaves pro-IL-1β and results in the release of interleukin-1β (IL-1β), a potent trigger of inflammation [4]. Another form of PCD, necroptosis serves as an intermediate form of cell death between necrosis and apoptosis capable of boosting antitumor immunity [5]. During this PCD, Z-DNA-binding protein 1 (ZPB1) induces a RIPK3/RIPK1 mediated necroptosis while inflammatory responses are also triggered [6]. In the cases of malignancies, downregulation of its components is considered as a poor prognostic sign [7].

Of all the possible PCD pathways, the one that will be triggered is induced by a combination of factors that include the harmful stimuli and the cell genome (the genes involved in a PCD must be intact) while multiple pathways can be triggered simultaneously. This is the case for ZnO nanoparticles (NPs) that can trigger apoptosis, pyroptosis and autophagy [8,9]. Moreover, the tumor microenvironment (TME) can also affect cell fate. For instance, it has been shown that non-small cell lung cancer causes an RNA-interference mediated suppression of GSDMD, which favors apoptosis and inhibits the pyroptotic cascade [10]. Similarly, miR-155-5p has been shown to be overexpressed in cases of osteosarcoma resulting in the inhibition of RIPK1, an essential component of necroptosis [11]. A second molecule, MLKL, that functions as the necroptotic executioner is also found to have reduced expression in multiple cancer types, including ovarian and gastric malignancies [11].

Although bimetallic nanoparticles (BNPs) and monometallic nanoparticles (MNPs) exhibit anticancer properties mostly by inducing apoptotic cell death [12], pyroptotic and necroptotic cell death have not been adequately studied. Thus, it is important to identify cancer types that are targeted via these alternative pathways triggered by BNPs, which have already shown efficacy in cells that can undergo apoptosis.

At the same time, when cell death occurs, biomolecules (the damage associated molecular patterns (DAMPs)) that serve as endogenous alarmins according to the danger theory are secreted/released. This heterogeneous group of molecules is produced by cells (including cancer cells) that experience injury or any kind of damage or oxidative stress and serve as potent stimuli of the sterile inflammatory response [13]. Their role in cancer is complex since current literature suggests a beneficial anti-cancer effect while, at the same time, a role in antitumor drug resistance is implicated [14]. The most important DAMPS include HMGB1 (high mobility group box 1) and HSP70 (heat-shock protein 70). HMGB1 functions as a danger alarmin that is secreted or passively released in the extracellular space, triggering inflammation. HSP70 has a similar role and is also released from damaged or necrotic tissue [15,16].

In our previous work, we showed that AgAu NPs not only inhibit tumor growth by triggering the apoptotic pathway but also prevent metastasis both in vitro (in a wound closure assay) and in vivo (using a mouse model) [17]. However, several questions remain to be answered, including whether PCD pathways other than apoptosis are induced and if danger molecules are released from the affected cancer cells. Here, we address these issues. We aimed to decipher the biological actions of these BNPs in different cancer cell lines. We investigated the expression of key genes for different cell death pathways and the secretion/release of IL1-β and of DAMPs in cell lines that were incubated with AgAu NPs, Ag NPs and Au NPs. These genes include *ZPB1*, *CXCL8*, *CCL20* and *CSF1* (for the investigation of the necroptotic pathway), *CASP1*, *IL-1β*, *IL-18*, *NLRP3* (for the pyroptotic pathway), *CASP3* and *BCL-2* (for apoptosis), *HSP70* (DAMP) and HMGB1 (DAMP). The cell lines used are the non-cancerous HEK293 and the cancerous MCF7, MDA-MB-231, LNCaP, C4-2B, SJ-GBM2, HCT116, and fibroblasts from Li-Fraumeni syndrome patients.

Our focus was to investigate two parameters of the anticancer effects of BNPs: firstly, the molecular pathways that are triggered (apoptosis, pyroptosis and necroptosis) in the cell lines tested; secondly, the effect of BNPs not only on the transcriptomic level but also on the secretion/release of IL-1β and HMGB1 by quantifying their extracellular levels. Collectively, these findings will allow researchers to identify the most sensitive cancer cells to BNP exposure and will give in-depth insight into their actions, allowing their effective and safe use in clinical applications.

## 2. Results

### 2.1. Characterization of NPs

Colloidal solutions of Ag and Au (monometallic and bimetallic) were formulated using tryptophan as a reducing and stabilizing agent. Absorption spectra of colloids includes typical bands for metal LSPR (localized surface plasmon resonance), and are shown in Figure 1A. For Ag, the maximum LSPR was observed at 415 nm while for Au the maximum LSPR was observed at 527 nm. Figure 1B shows the samples before and after dilution.

The analysis of metal content in bimetallic AgAu NPs was performed with the XPS method (Figure 1C,D) based on the energy levels of electrons, namely 374 (3d3/2) and 368 (3d5/2) eV for Ag and 88 (4f5/2) and 84 (4f7/2) eV for Au [18]. For the AgAu sample the values were 373.9, 367.9, 87.5 and 83.9 respectively. Due to the XPS data, according to the calculated ratio of the intensities of the gold and silver bands, the content of metals in bimetallic NPs corresponds to that used in the synthesis (Ag:Au = 3:1). The localized position at λ LSPR max = 415 (Ag), 475 (AgAu(3:1)) and 528 (Au) nm. The size and morphology, as revealed by TEM and SEM images, is shown in Figure 2. Moreover, the colloidal solutions show a long-term stability, while compared to the analogy of Metal:Tryptophan 1:1, the analogy tested in this study (the double molar excess of tryptophan, 1:2) shows increased aggregation [12].

### 2.2. Cytotoxicity Studies

Based on the MTS assay results (Figure 3, Table 1), BNPs show a dose-dependent toxicity. Of all NPs tested, the lowest decrease in viability is observed for Au NPs. The highest decrease in viability was observed for the HCT116 cell line (approximately 10% regardless of the NP type for the concentration of 50 μg/mL) as has been shown in our previous work [12] and in the present study too (shown in Figure 3G). The decrease in viability is lower in the case of the LNCaP cell line in the presence of Ag NPs (approximately 60%, 50 μg/mL) and in MCF-7 cells (viability that barely exceeded 60% for 50 μg/mL, AgAu NPs) followed by MDA-MB-231 cells (approximately 65% for both Ag NPs and AgAu NPs, 50 μg/mL). C4-2B and SJ-GBM2 cell lines experienced an even weaker cytotoxicity as shown by the decrease in viability; all concentrations and all NPs tested led to a viability that did not drop below 80%. Fibroblasts from Li-Fraumeni syndrome patients were proven to be the most tolerant to the action of NPs showing an unchanged viability (100%) for all NP types and for all concentrations tested. The experiment was repeated three times in triplicate with similar results. Figure 3 and Table 1 show the exact values of the experiments.

### 2.3. Cell Death Pathways Induction

Based on our results, incubation with 50 μg/mL of Ag, Au or AgAu NPs resulted in the upregulation of *CASP1*, which serves as an indicator of pyroptosis, most notably in MDA-MB-231 (all three NPs tested) and LNCaP (Au and AgAu NPs) cell lines as shown in Figure 3A. The highest downregulation (approximately 7 times) was noted in the case of HCT116 (AgAu NPs). Regarding the other genes whose expression was used as a pyroptotic marker (IL-1β, IL-18 and NLRP3), the most notable change was noted for IL-1β. As shown in Table 2, the highest upregulation of IL-1β was observed for MDA-MB-231, LNCaP and HCT116 (2.83, 2.64 and 2.29 respectively) after incubation with AgAu NPs. *NRLP3* and *IL-18* did not show any significant upregulation except for Ag NPs that showed an upregulation (2.29-fold change) of NLRP3 in the MDA-MB-231 cell line.

CASP3, where its upregulation was considered as an indicator of apoptosis, was upregulated mainly in MDA-MB-231 and HCT116 cell lines. On the contrary, *CASP3* was downregulated (approximately by a fold change of 5 times) in the SJ-BM2 cell line (Ag, Au NPs). Moreover, the anti-apoptotic *BCL-2* was found to be down-regulated in both HCT116 and MDA-MB-231 cells (by −1.9 and −2.38 for AgAu NPs respectively) while it remained practically unchanged for all other cell lines.

ZPB1 expression, where its upregulation was considered as an indicator of necroptosis, was upregulated mainly in the MDA-MB-231 cell line for all NPs tested. The highest upregulation was observed after the incubation with Ag NPs (3 times fold change). For most cell lines (Li-Fraumeni fibroblasts, LNCaP, C4-2B, SJ-GBM2) ZPB1 was downregulated as shown in Table 2. Regarding *CXCL8* and *CCL20*, no significant upregulation was noticed for any cell lines. *CSF1* showed a slight upregulation (1.74-fold change) in the MDA-MB-231 (after incubation with AgAu NPs as shown in Table 2). On the contrary, the *CSF1* mRNA showed a −5.5 downregulation for the Li-Fraumeni fibroblasts after Ag NP incubation.

Regarding the expression of DAMPS, HMGB1 was only upregulated in the case of HEK293 (AgAu NPs) and C4-2B (AgAu NPs) and *HSP70* was mainly downregulated for all cell lines and most notably in Li-Fraumeni fibroblasts, C4-2B and SJ-GBM2 cell lines. All the values of fold change are shown in Table 2 in Arbitrary Units (AU), which represent the expression of each gene’s mRNA normalized to the mRNA of GAPDH.

### 2.4. Extracellular DAMP Levels

As indicated by the extracellular levels of HMGB1 (Figure 4), the most notable difference was detected in HCT116 cells. Incubation with Ag NPs, Au NPs, and AgAu NPs led to an increase of approximately 60% (compared to untreated cells) of extracellular HMGB1 levels. A more subtle increase was observed for MDA-MB-231 and, MCF-7 cells (almost 10% for AgAu NPs). The non-cancerous cell line HEK293 and the rest of the cancerous cell lines did not show altered levels following NP exposure.

### 2.5. Extracellular IL-1β Levels

The levels of IL-1β after incubation of MDA-MB-231 with Ag and AgAu NPs were 7.9 and 12.8 pg/mL, respectively. Regarding the LNCaP cell line, incubation with Ag and AgAu NPs resulted in 6.3 and 19.2 pg/mL of IL-1β as shown in Figure 5. In every other cell line and conditions (Ag, Au and AgAu NPs and no NP treatment), no extracellular IL-1β was detected.

## 3. Discussion

Cancer cell death is the goal of any drug purposed for malignancy treatment [20]. Among the different ways that this can be achieved, programmed cell death (PCD) is preferred over necrosis, due to the uncontrolled process and the cell content leakage of the latter [21]. Regarding PCD pathways, the apoptotic pathway is highly desirable due to the controlled and regulated manner of the process [22], while at the same time it is the most studied for several anti-tumor treatments [1] including NPs [23,24]. However, cancer cells can be apoptosis resistant, and several novel drugs target alternative PCD pathways [2].

In terms of cytotoxicity, our findings support those of our previous study [12] since the tested NPs showed a dose dependent toxicity. However, a large heterogeneity in the toxicity is observed. For instance, MCF-7 cells had an approximately 40% decrease in their viability, in contrast to the fibroblasts that derived from Li-Fraumeni patients that experienced nearly no toxicity (Figure 3). It is important to clarify the PCD mechanism involved, to decipher these results and properly evaluate the effects of AgAu NPs.

Pyroptosis is a lytic type of PCD characterized by cell swelling and proinflammatory molecule release [25]. While like apoptosis in its features, its morphological characteristics are distinct. The trigger of this pathway is the induction of CASP1 that is mediated via a harmful stimulus [26]. Although linked to infection, pyroptotic cell death arises as a promising strategy for novel anti-cancer treatments [27,28]. Indeed, recent studies show that metallic, inorganic, and biomimetic NPs can exhibit anticancer effects via pyroptosis [29,30]. This PCD pathway is linked to tumor suppression properties mostly mediated via the reprogramming of the tumor microenvironment [31]. Interestingly, the mRNA fold change of *CASP1* gene shows that pyroptosis is triggered on the P53 deficient MDA-MB-231 cells as well as on LNCaP cancer cells (Table 2). It should be noted that *NLRP3* gene (a complex protein that senses danger signals) is also upregulated in MDA-MB-231 cells that were incubated with AgNPs. Cisplatin has been found to also trigger the *CASP1/NLRP3* pyroptotic pathway in MDA-MB-231 cells [32]. Moreover, *IL-1β* mRNA levels (Table 2) and extracellular IL-1β (Figure 5) show an increase for both MDA-MB-231 and LNCaP cells treated with Ag and AgAu NPS, a finding that further confirms that pyroptosis is triggered and that inflammatory molecules (IL-1β) are secreted/released. Thus, AgAu NPs are also capable of triggering the distinct pyroptotic pathway that arises as a major target for novel treatments.

Interestingly, HCT116 showed an increase of IL-1β mRNA but not in the extracellular IL-1β. However, the key characteristics of pyroptosis include CASP1 increase for the canonical pathway and the secretion/release of IL-1β for the canonical and the *CASP1*-independent pathway [4]. Since extracellular IL-1β is not detected and *CASP1* is not upregulated, pyroptosis is rather unlikely for HCT116. Thus, apoptosis remains the triggered cell death pathway for HCT116 cells as suggested by the mRNA changes of *CASP3* and *BCL-2*, which agrees with our previous study [12].

Necroptosis, the second type of PCD that we examined is a form of necrosis that is regulated and triggered by death receptors [33]. This PCD pathway can be triggered by ZPB1 (Z-DNA-binding protein 1) that, similarly to pyroptosis, was linked to infection [6,34]. Unlike pyroptosis, necroptosis is a secondary, caspase-independent PCD type [35]. However, it remains far less studied than apoptosis and pyroptosis [33], although research suggests that immunotherapy that triggers necroptosis holds promise [36]. Quite recently [37], AgNPs were shown to trigger mixed types of programmed cell death (including necroptosis) in pancreatic carcinoma. Other types of metal NPs have also been shown to trigger the necroptotic pathway [38,39]. Based on mRNA fold change only MDA-MB-231 cells show a triggering of the ZPB1-dependent necroptotic pathway, while CSF1, a proinflammatory cytokine whose expression increases during necroptosis [40], shows a mild upregulation after AgAu NPs incubation (1.74 as shown in Table 2).

Unlike the P53 deficient MDA-MB-231 cells, both SJ-GBM2 and Li-Fraumeni fibroblasts (that are both P53 deficient) showed tolerance to the cytotoxic effects of all NPs tested. Although P53 independent pathways can lead to apoptosis (as suggested via the activation of CASP3 in other studies [17,41], this does not occur in these cell lines (*CASP3* expression is downregulated as shown in Table 2). Moreover, a 2018 study [42] showed that irradiation of SJ-GBM2 cells not only leads to radioresistance but also promotes a more aggressive phenotype (higher proliferation rates) and concluded that an alternative to radiation is needed. Unfortunately, AgAu NPs also fail to act as anticancer agents for glioblastoma cells.

In total, MDA-MB-231 cells experience a mixed PCD pathway that involves pyroptosis, necroptosis and apoptosis (as indicated by the downregulation of *BCL-2* and the *CASP3* upregulation) while IL-1β is also secreted/released. To the best of our knowledge, this is the first article to show that AgAu alloys can have antitumor effects by triggering mixed PCD pathways. Moreover, the necroptotic pathway offers significant advantages since, except for being an apoptosis-alternative, it can also boost the innate immunity and thus exhibit anti-cancer effects and inhibition of tumor progression [5,43].

Regarding DAMPs, we tested the expression of two important types, HMGB1 and HSP70 (Table 2). The former has a complex and sometimes paradoxical role, since studies suggest different actions: danger signaling [15], promoting tumor cell proliferation and tumor cell death [44,45]. HSP70 has also multiple roles including apoptosis suppression in malignancies [46].

Based on our results, the most notable upregulation for HMGB1 was noted for HEK293, C4-2B and HCT116 (after incubation with AgAu NPs), while no notable up-regulation was observed for HSP70 in any of the cell lines tested. Furthermore, the extracellular HMGB1 (Figure 4) showed its higher value (compared to untreated cells) for HCT116 cells. This finding is in accordance with the apoptotic death of HCT116 cells via the P53/BAX/BCL pathway (as shown in our previous study [12]) since apoptotic cells release DAMPS [47].

## 4. Materials and Methods

### 4.1. Nanoparticle Preparation and Characterization

Colloids of monometallic and bimetallic Ag and Au NPs were prepared through chemical reduction of silver nitrate (AgNO_3_) and tetra-chloroauric acid (HAuCl_4_) with tryptophan as previously described [9,12]. The concentration of metal in the prepared solutions was C(M) = 10^−4^ mol/L and the ratio of components for all colloids was v(M):v(Trp) = 1:2. The molar ratio of AgAu nanoparticles used was Ag3Au1. The initial tryptophan solutions were adjusted to a pH value of 10 with 1 N of NaOH that was heated until the point of boiling and injection of AgNO_3_/HAuCl_4_ followed.

### 4.2. Cell Culture

The following cell lines were used: HEK293 (a non-cancerous cell line of epithelial origin), MFC7 (human breast cancer), MDA-MB-231 (human breast cancer), LNCaP (human prostate cancer), C4-2B (human prostate cancer), SJ-GBM2 (human glioblastoma), HCT 116 (human colorectal cancer) and fibroblasts derived from Li-Fraumeni patients. MDA-MB-231, Li-Fraumeni fibroblasts and SJ-GBM2 cell lines are p53 deficient [48,49]. Cells were grown on suitable medium:for HEK293, HCT 116, MDA-MB-231, MCF-7 and Li-Fraumeni fibroblasts DMEM High Glucose (BioSera, Shanghai, China) was used with 10% FBS (PAN Biotech, Aidenbach, Germany), 100 U/mL penicillin and 100 g/mL streptomycinfor LNCaP RPMI (Gibco, Waltham, MA, USA) was used with 10% FBS and 100 U/mL penicillin and 100 g/mL streptomycinfor SJ-GBM2 IMDM 1× (PAN BIOTECH) containing stable Glutamine 25 mM, HEPES (w: 3.024 g/: NaHCO_3_) with ITS (AOF ITS Supplement, Millipore, Burlingtone, MA, USA) and 20% FBS at 37 °C until reaching 70% confluency.

ATCC supplied HEK293 (ACRL-1573™), HCT116 (CCL-247™), MDA-MB-231 (HTB-26™), MCF7 (HTB-22™), LNCaP (CRL-1740™), C4-2B (CRL-3315™) cell lines, Department of Tumor Biology and Department of Genitourinary Medical Oncology, Division of Cancer Medicine, The University of Texas, M. D. Anderson Cancer Center, Houston, USA supplied Li-Fraumeni fibroblasts and Cancer Center, Department of Cell Biology and Biochemistry and Texas Tech University Health Sciences Center supplied SJ-GBM2 cell lines.

### 4.3. Cytotoxicity Assay (MTS Assay)

For the cytotoxicity assay (MTS assay, CellTite96, Aqueous One Solution, Promega, Madison, WI, USA), cells were seeded in a 96-well plate, and they were incubated (23 h) with AgAu NPs, Ag NPs and Au NPs in 3 concentrations (30, 40 and 50 μg/mL). Regarding BNPs, 30 μg/mL contain 11 μg/mL of Au and 19 μg/mL of Ag, 40 μg/mL of BNPs contain 15 μg/mL of Au and 25 μg/mL of Ag, and 50μg/mL of BNPs contain 11 μg/mL Au and 19 μg/mL Au. Three types of control were used: a positive (cells with culture medium only and no BNP incubation), a negative (BNPs without cells) and a background control (nutrient medium alone). Each well contained 5000 cells/well. In brief, after incubation (approximately 3 h with the MTS solution), formazan was produced by NADP(H) dependent dehydrogenases. Absorbance was measured at 490 nm. Viability was calculated as a % ratio of treated cells to untreated cells. Prior to this the blank wells absorbance was subtracted from all wells. The detailed process has been previously described [12].

### 4.4. RNA Extraction, cDNA Synthesis and Real-Time PCR

For the RNA extraction, 6-well plates were seeded with 5 × 10^4^ cells. Cells of all cell lines mentioned were incubated (23 h) with 50 μg/mL of AgAu NPs, Ag NPs and Au NPs (each well contained only one type of NP). After the end of incubation, nutrient medium was removed, and the RNA extraction was performed using NucleoZOL (Macherey-Nagel, Düren, Germany). Nutrient medium was removed, and the RNA extraction was performed using NucleoZOL (Macherey-Nagel, Düren, Germany). cDNA synthesis was performed using PrimeScript First-Strand cDNA kit (Takara Bio Europe SAS, Saint-Germain-en-Laye France). RNA (1 μg) was incubated for 30 min at 37 °C followed by 5 s incubation at 85 °C in a reaction containing 500 μg of Oligo dT, 10 mM deoxyribonucleotide triphosphates 5X first-strand buffer 0.1 M dithiothreitol and 200 U/mL reverse transcriptase. Subsequently, mRNA fold change was estimated using GAPDH as a gene of reference. The studied mRNAs correspond to *ZPB1*, *CASP1*, *CASP3*, *BCL-2*, *HSP70*, *HMB1*, *CXCL8*, *CSF1*, *CCL20*, *NLRP3*, *IL-1β* and, *IL-18* genes (Genes Of Interest, GOI). Quantitive real-time PCR was per-formed on an ABI Prism apparatus (Applied Biosystems, Foster City, CA, USA). Gene expression levels were normalized by subtracting Ct value of the GAPDH RNA from that of GOI using the equation (ΔCt = −|CtGOI − CtGAPDH|). Relative expression of *ZPB1*, *CASP1*, *CASP3*, *BCL-2*, *HSP70*, *HMGB1*, *CXCL8*, *CSF1*, *CXCL20*, *NLRP3*, *IL-1β* and, *IL-18* was determined comparing the samples from cells that were incubated with NPs with untreated cells, using the 2ΔΔCt model, in which ΔΔCt = ΔCt_GOI_ − ΔCt_GAPDH_. All samples were held in duplicate to ensure reproductivity. The primers used for these reactions are shown in Table 1 and Table 3. Primer sequences were used in the Real Time PCR. All cell lines were incubated for 23 h with 50 μg/mL of AgAu NPs, Ag NPs and Au NPs and the nutrient, cell-free medium was collected. Extracellular levels of HMBG1 and IL-1β were quantified using the ELISA technique (Human HMGB1 ELISA kit by FineTest, Wuhan and Human Interleukin 1β, IL-1β ELISA Kit by Cusabio). All experiments were performed in duplicate, following the manufacturer’s instructions.

### 4.5. Evaluation of Extracellular DAMPs (HMGB1) and IL-1β

All cell lines were incubated for 23 h with 50 μg/mL of AgAu NPs, Ag NPs and Au NPs and the nutrient, cell-free medium was collected. Extracellular levels of HMBG1 and IL-1β were quantified using the ELISA technique (Human HMGB1 ELISA kit by FineTest, Wuhan and Human Interleukin 1β, IL-1β ELISA Kit by Cusabio, Houston, TX, USA). All experiments were performed in duplicate, following the manufacturer’s instructions.

### 4.6. Statistical Analysis

Statistically significant differences between the values of the samples were evaluated by one-way analysis of variance (ANOVA) as well as the nonparametric Kruskal–Wallis method using GraphPad version 3.00 (GraphPad Software, San Diego, CA, USA). *p* < 0.05 value was considered as statistically significant.

## 5. Conclusions

Our study shows that AgAu NPs, as well as Ag NPs and Au NPs have anticancer effects, not only via triggering the apoptotic pathway but also by triggering pyroptosis and necroptosis and even mixed PCD pathways (MDA-MB-231). Moreover, AgAu NPs result in the release of IL-1β for MDA-MB-231 and LNCaP cell lines and thus extracellular IL-1β levels could have a role in the prediction of response to pyroptosis-inducing treatments. However, it needs to be answered whether there can be a prediction model to show in advance which pathway will be triggered. P53 deficiency alone is not enough, since P53 deficient cells can undergo P53 independent apoptosis [12]. Furthermore, the high levels of extracellular HMGB1 in HCT116 cells (which experience high cytotoxicity after AgAu NPs) is a promising finding, suggesting a possible use of HMGB1 level as a means of early detection of response to treatment, similar to IL-1β for pyroptosis. However, further studies that investigate the connection between AgAu treatment response of tumors and the release of either HMGB1 or IL-1β are required to test this hypothesis. Finally, both necroptosis and pyroptosis are novel targets for cancer treatments and their exact effects need to be further elucidated. Taken together, AgAu NPs show antitumor effects in different cancer cell lines with potential applications in a variety of genetically heterogenous cancers.

## Figures and Tables

**Figure 1 cancers-14-01546-f001:**
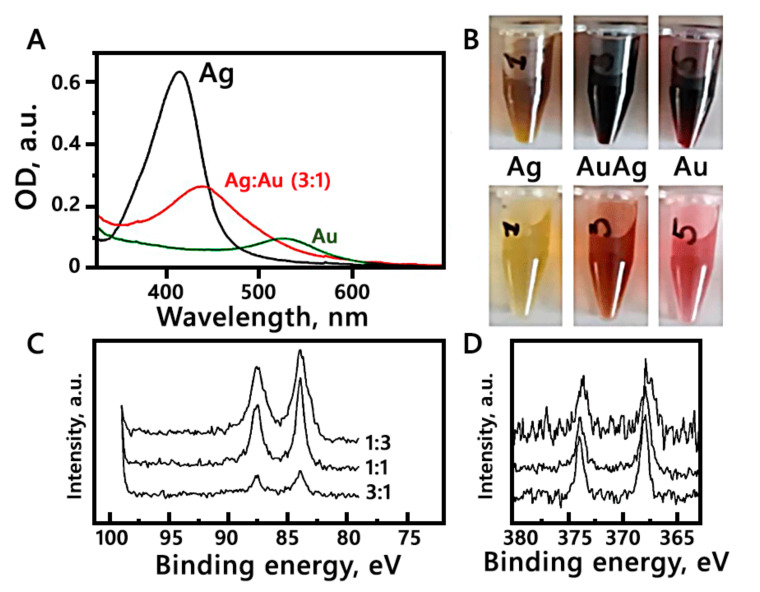
Panel (**A**) shows absorption spectrum of Ag, Au and AgAu NPs. Panel (**B**) shows Ag, Au and AgAu samples before (upper part) and after (lower part) dilution. Panels (**C**,**D**) show high-resolution XPS spectra of the series of bimetallic AgAu colloids, (AgAu(3:1), which was used in this study, AgAu(1:1) and AgAu(1:3)) in the range of Au and Ag (**D**).

**Figure 2 cancers-14-01546-f002:**
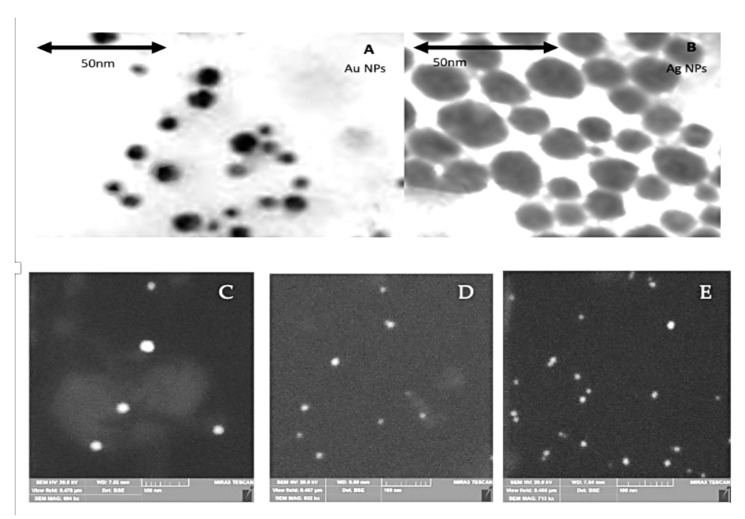
Electron microscopy images. Panels (**A**,**B**) show TEM images of Au and Ag NPs respectively. Panels (**C**–**E**) show SEM images of Ag, AgAu and Au NPs respectively. Panels (**C**–**E**) are reprinted with permission [19].

**Figure 3 cancers-14-01546-f003:**
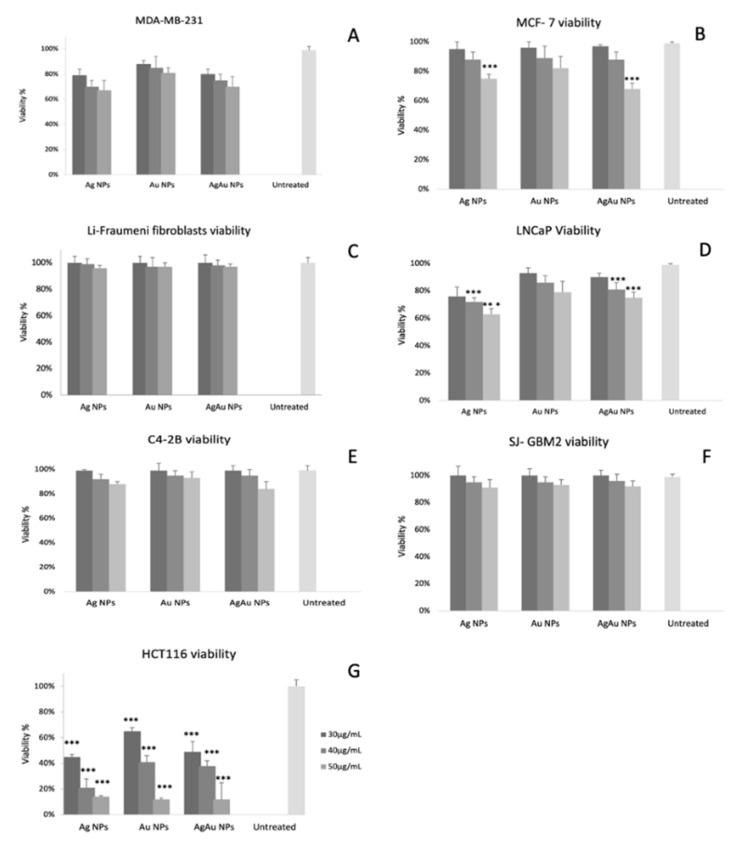
Viability percentage of all cancer cell lines (MDA-MB-231, MCF-7, fibroblasts derived from Li-Fraumeni patients, LNCaP, C4-2B, SJ-GBM2 and HCT116) in panels (**A**–**G**) respectively, incubated with NPs (Ag NPs, Au NPs, AgAu NPs) for 23 h. The *** symbol shows statistical significance using one way ANOVA (*p* < 0.05) compared to untreated cells.

**Figure 4 cancers-14-01546-f004:**
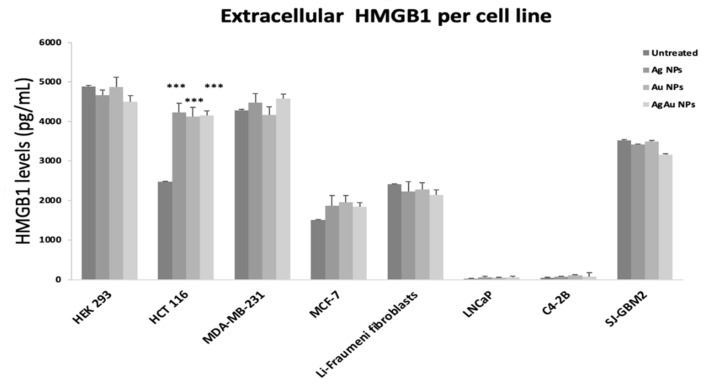
Extracellular levels of HMGB1 per cell line, after incubation with 50 μg/mL of Ag, Au or AgAu NPs. The *** symbol shows statistical significance using one way ANOVA (*p* < 0.05) compared to untreated cells.

**Figure 5 cancers-14-01546-f005:**
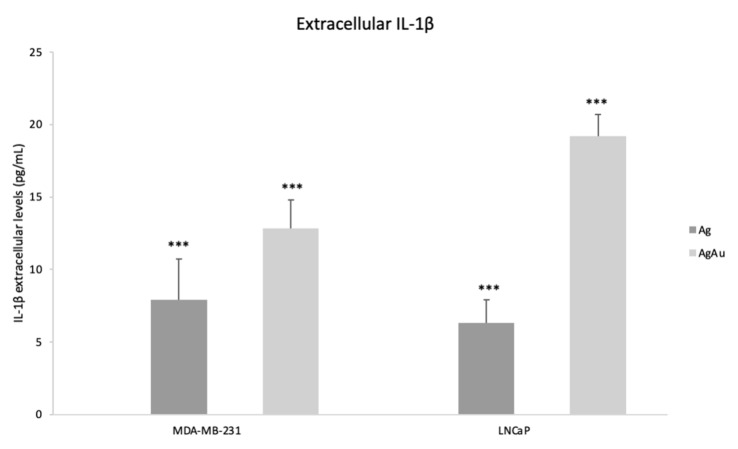
Extracellular levels of IL-1β per cell line, after incubation with 50 μg/mL of Ag, AgAu of MDA-MB-231 and LNCaP cells. The *** symbol shows statistical significance using one way ANOVA (*p* < 0.05) compared to untreated cells. In every other cell line and conditions (Ag, Au and AgAu NPs and no NP treatment), no extracellular IL-1β was detected.

**Table 1 cancers-14-01546-t001:** Viability percentage ± SD between two experiments of cells incubated with Ag, Au, AgAu NPs (30, 40 and 50 μg/mL) and untreated cells. The asterisk symbol (***) shows statistical significance using one way ANOVA (*p* < 0.05) compared to untreated cells.

Viability of MDA-MB-231 (%)	Ag NPs	Au NPs	AgAu NPs	Untreated
30 μg/mL	80 ± 5	88 ± 3	81 ± 4	
40 μg/mL	75 ± 5	85 ± 9	70 ± 5	99 ± 3
50 μg/mL	70 ± 8	81 ± 4	67 ± 8	
**Viability of MCF-7 (%)**				
30 μg/mL	95 ± 6	96 ± 6	97 ± 5	
40 μg/mL	88 ± 5	89 ± 8	88 ± 5	99 ± 1
50 μg/mL	75 ± 3 ***	82 ± 5	68 ± 4 ***	
**Viability of Li-Fraumeni fibroblasts (%)**				
30 μg/mL	100 ± 4	100 ± 2	100 ± 2	
40 μg/mL	99 ± 2	97 ± 3	98 ± 1	100 ± 4
50 μg/mL	96 ± 3	97 ± 2	97 ± 1	
**Viability LNCaP (%)**				
30 μg/mL	76 ± 7	93 ± 4	90 ± 3	
40 μg/mL	72 ± 3 ***	86 ± 5	81 ± 5 ***	99 ± 1
50 μg/mL	63 ± 4 ***	79 ± 8	75 ± 4 ***	
**Viability of C4-2B (%)**				
30 μg/mL	100 ± 1	100 ± 6	100 ± 4	
40 μg/mL	92 ± 6	95 ± 4	95 ± 5	99 ± 4
50 μg/mL	88 ± 4	93 ± 5	84 ± 6	
**Viability of SJ-GBM2 (%)**				
30 μg/mL	100 ± 7	100 ± 5	100 ± 4	
40 μg/mL	95 ± 4	95 ± 4	96 ± 5	99 ± 2
50 μg/mL	91 ± 6	93 ± 4	92 ± 4	
**Viability of HCT116 (%)**				
30 μg/mL	45 ± 2 ***	65 ± 3 ***	49 ± 8 ***	
40 μg/mL	21 ± 7 ***	41 ± 5 ***	38 ± 4 ***	100 ± 5
50 μg/mL	14 ± 1 ***	12 ± 1 ***	12 ± 13 ***	

**Table 2 cancers-14-01546-t002:** mRNA fold changes in AU after incubation with Ag NPs (**A**). Au NPs (**B**) and AgAu NPs (**C**). The asterisk (***) shows statistical significance (*p* < 0.05) compared to untreated cells.

**A** **(Ag NPs)**	**HEK293**	**MDA-MB-231**	**MCF-7**	**Li-Fraumeni Fibroblasts**	**LNCaP**	**C4-2B**	**SJ-GBM2**	**HCT116**
** *CASP1* **	−1.11	4.92	−1.62	−1.23	1.74	−1.63	−2.29	−1.58
** *CASP3* **	1.22	2.00	−1.62	−1.14	−1.23	−1.74	−6.09 ***	1.50
** *BCL-2* **	−1.10	−2.86	−1.23	1.00	1.00	−1.33	1.36	−1.80
** *ZPB1* **	−1.11	3.60	−1.32	−3.57	−2.46	−1.86	−3.03	1.00
** *HMGB1* **	−1.23	1.27	−1.41	−1.23	1.14	−1.14	−2.83	1.68
** *HSP70* **	1.00	−1.75	−1.51	1.14	1.31	−4.00	−1.51	1.23
** *CXCL8* **	1.07	1.15	1.15	−7.60	−1.20	−1.53	−1.24	1.07
** *CSF1* **	1.23	1.40	1.62	−5.5	−2.50	1.20	−1.33	−1.14
** *CCL20* **	−1.23	1.10	1.31	1.15	−1.15	1.23	1.50	1.23
** *NLRP3* **	1.14	2.29	1.30	−1.07	1.30	1.07	1.15	−1.75
** *IL-1β* **	1.62	2.46	1.07	−1.07	2.82	1.23	1.23	1.87
** *IL-18* **	1.75	1.15	1.86	1.31	1.00	1.50	1.10	1.23
**B** **(Au NPs)**	** *HEK293* **	**MDA-MB-231**	**MCF-7**	**Li-Fraumeni Fibroblasts**	**LNCaP**	**C4-2B**	**SJ-GBM2**	**HCT116**
** *CASP1* **	−1.33	2.14	−4.92	1.07	6.96 ***	−2.29	−2.00	1.31
** *CASP3* **	1.00	2.14	−1.41	1.00	−1.20	−2.29	−6.08 ***	1.70
** *BCL-2* **	1.01	−1.78	1.62	−1.33	−1.13	1.36	1.10	−1.31
** *ZPB1* **	2.21	1.93	1.31	−1.41	−1.74	−2.00	−3.48	1.56
** *HMGB1* **	−1.20	1.14	−1.51	1.00	1.74	1.40	−1.86	2.21
** *HSP70* **	−1.47	1.18	1.31	−3.03	1.30	−1.86	−2.83	1.31
** *CXCL8* **	1.31	−1.75	−1.88	−2.17	1.31	1.86	−1.66	1.62
** *CSF1* **	1.29	1.15	−2.17	−1.33	−1.64	−1.25	1.10	1.23
** *CCL20* **	1.03	1.74	−2.86	−1.53	1.41	−1.43	1.10	1.10
** *NLRP3* **	1.42	1.86	1.62	−1.54	−1.42	1.00	−1.15	1.74
** *IL-1β* **	1.41	1.23	1.23	1.23	1.60	1.39	−1.43	1.41
** *IL-18* **	1.23	1.41	1.32	−1.07	1.00	1.30	−1.66	1.87
**C** **(AgAu NPs)**	** *HEK293* **	**MDA-MB-231**	**MCF-7**	**Li-Fraumeni Fibroblasts**	**LNCaP**	**C4-2B**	**SJ-GBM2**	**HCT116**
** *CASP1* **	2.46	3.03	1.51	−1.75	6.96 ***	−1.33	1.14	−6.66 ***
** *CASP3* **	1.00	2.07	1.07	−1.75	−1.23	−1.07	−2.46	2.00
** *BCL-2* **	1.12	−2.38	1.40	−1.30	1.00	−1.60	1.80	−1.90
** *ZPB1* **	−1.11	3.03	−2.00	−2.29	−2.29	−2.29	−2.64	1.14
** *HMGB1* **	2.46	1.27	−1.07	1.23	1.74	3.24	−1.41	−1.00
** *HSP70* **	−1.96	−1.36	−1.07	−3.03	−1.51	−2.46	−2.00	−1.19
** *CXCL8* **	−1.33	1.32	1.32	−1.08	−1.15	−1.15	1.50	1.00
** *CSF1* **	1.23	1.74	−1.42	−1.23	1.10	1.41	−2.00	1.10
** *CCL20* **	1.10	1.00	1.75	1.22	1.41	1.23	1.29	1.62
** *NLRP3* **	1.08	1.63	1.29	1.00	1.00	−2.00	1.11	−2.70
** *IL-1β* **	1.14	2.83	1.75	1.00	2.64	1.32	−1.10	2.29
** *IL-18* **	1.75	1.74	−1.53	−1.89	−1.43	1.67	1.15	1.32

**Table 3 cancers-14-01546-t003:** Primer sequences.

Gene	Forward Primer	Reverse Primer
ZPB1	5′-TGGTCATCGCCCAAGCACTG-3′	5′-GGCGGTAAATCGTCCATGCT-3′
CASP1	5′-GCCTGTTCCTGTGATGTGGAG-3′	5′-TGCCCACAGACATTCATACAGTTTC-3′
CASP3	5′-TGGTTCATCCAGTCGCTTTG-3′	5′-CATTCTGTTGCCACCTTTCG-3′
BCL-2	5′-GATGTGGATGCCTCTGCGAAG-3′	5′-CTGCTGATGTCTCTGGATCT-3′
HSP70	5′-ATGTCGGTGGTGGGCATAGA-3′	5′-CACAGCGACGTAGCAGCTCT-3′
HMGB1	5′-ATATGGCAAAGCGGACAAG-3′	5′-AGGCCAGGATGTTCTCCTTT-3′
CXCL8	5′-CAGTTTGCCAAGGAGTGCT-3′	5′-ACTTCTCCACAACCCTCTGC-3′
CSF1	5′-TGGCGAGCAGGAGTATCAC-3′	5′-AGGTCTCCATCTGACTGTCAAT-3′
CCL20	5′-GTGCTGCTACTCCACCTCTG-3′	5′-GCATTGATGTCACAGCCTTCA-3′
NLRP3	5′-TTCAATGGCGAGGAGAAGGC-3′	5′-ACGTGTCATTCCACTCTGGC-3′
IL-1β	5′-CCTTGTCGAGAATGGGCAGT-3′	5′-TCCTGTCGACAATGCTGCCT-3′
IL-18	5′-TCTTCATTGACCAAGGAAATCGG-3′	5′-TCCGGGGTGCATTATCTCTAC-3′
GAPDH	5′-CATCTCTGCCCCCTCTGCTG-3′	5′-GCCTGCTTCACCACCTTGTTG-3′

## Data Availability

The data generated and/or analyzed during the current study are available from the corresponding author on reasonable request.

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
