# Peer review of "Ag/Au Bimetallic Nanoparticles Trigger Different Cell Death Pathways and Affect Damage Associated Molecular Pattern Release in Human Cell Lines"

_cancers, 2022, doi:10.3390/cancers14061546_

Round 1
Reviewer 1 Report
Please see attachment

Reviewer 2 Report
The article “Ag/Au bimetallic nanoparticles trigger different cell death pathways and affect Damage Associated Molecular Patterns release in human cell lines”, submitted to Cancers, addresses the induction of apoptosis, necrosis, or pyroptosis in different cancer cell lines after the treatment with AgAu NPs, Ag NPs, and Au NPs. The authors claim that NPs possess strong cytotoxic activity and trigger different types of programmed cell death in breast, prostate, and colorectal cancer. However, some points need to be clarified.
- The article addresses the investigation of types of programmed cell death, triggered with AgAu NPs, Ag NPs and Au NPs. However, conclusions about the apoptosis induction are made based only on the caspase-3 expression, which is not enough. It was a good idea to add IL-1b investigation to the previous article version to investigate pyroptosis. Additional tests, such as Annexin V/Propidium iodide staining to determine apoptosis/ necrosis would strengthen the conclusions about the type of triggered cell death.
- The statements added to the "conclusion" section are questionable. Why do the authors claim that "IL-1β levels could have a role in the prediction of response to pyroptosis-inducing treatments"? In the current investigation IL-1β was used as a pyroptosis marker to determine the type of programmed cell death. To conclude about its prognostic value some in vivo studies should be performed with calculating correlations between viability/ dynamics of tumor growth and the IL-1β levels. Same question arises regarding HMGB1, which was used initially as the evidence of DAMP release, but the conclusion is made vise verse about its prognostic value. So, cause and effect relationships need to be clarified
Minor comment:
Figure 6 should be centered related to the text.
Round 2
Reviewer 1 Report
Authors made considerable job in collecting a lot of data. I believe that this can be of some use in the future researches. I wish the authors all the best and hope that they will plan their future articles more thoroughly, especially graphical content.
Reviewer 2 Report
Dear authors,
I am sorry to hear that your colleagues were impacted by the events in Ukraine. I suppose that the addition of data about the downregulation of Bcl-2 gene expression provides a good confirmation of apoptosis induction. The article now can be accepted.